# Antagonism in Orthotospoviruses Is Reflected in Plant Small RNA Profile

**DOI:** 10.3390/v17060789

**Published:** 2025-05-30

**Authors:** Md Tariqul Islam, Kaixi Zhao, Nathan Johnson, Michael Axtell, Cristina Rosa

**Affiliations:** 1Department of Plant Pathology and Environmental Microbiology, The Pennsylvania State University, University Park, PA 16802, USA; mki5039@psu.edu (M.T.I.); kzhao5699@gmail.com (K.Z.); 2Department of Biology, The Pennsylvania State University, University Park, PA 16802, USA; jax523@gmail.com (N.J.); mja18@psu.edu (M.A.); 3Huck Institutes of the Life Sciences, The Pennsylvania State University, University Park, PA 16802, USA; 4Millennium Science Initiative–Millennium Institute for Integrative Biology (iBio), Santiago 8331150, Chile; 5Centro de Genómica y Bioinformática, Facultad de Ciencias, Universidad Mayor, Santiago 8580000, Chile

**Keywords:** mixed infection, virus–virus interaction, TSWV, INSV, super infection exclusion, RNAi

## Abstract

Mixed infections of plant viruses are commonly found in natural patho-systems and present a valuable opportunity to understand how multiple viruses can co-infect the same host. Tomato spotted wilt orthotospovirus (TSWV) and impatiens necrotic spot orthotospovirus (INSV) are present in the same geographic areas and are closely related. More mixed infections of TSWV and INSV have been reported in recent years, and the INSV host range has been reported to be increasing. In a previous study, we isolated and characterized one strain of INSV and one of TSWV and found that they have an antagonistic relationship in their vectors. However, we were unable to determine whether this antagonism extends to the host plant or to uncover the underlying mechanisms and the host’s contribution. Here, we show that TSWV and INSV exhibit antagonistic interactions in the host plant, as evidenced by a lower viral titer in mixed infections compared to single infections. Using small RNA sequencing, we identified that the host plant contributes to this antagonism through differential small RNA processing, which appears to regulate viral replication and the success of infection. This research advances our understanding of virus–virus and virus-host interactions and presents opportunities for leveraging these dynamics in integrated pest management strategies.

## 1. Introduction

Mixed infections of plant viruses are common in nature and can influence disease severity, transmission efficiency, and host responses [1,2,3,4,5]. These interactions can be synergistic, where co-infection enhances viral replication and pathogenicity, or antagonistic, where one virus suppresses the other [1,2,3,4,5]. Unrelated viruses with a few exceptions interact with each other in a synergistic way. Whereas, for related viruses, the interactions are mostly antagonistic. These antagonistic relationships, also called cross-protection or super-infection exclusion [6], exclude a virus from tissues already infected by another virus. This phenomenon can be used as a form of cross-protection in agriculture, where mild virus strains are used to ‘vaccinate’ plants against severe strains of the same virus [7]. Super-infection exclusion has often been attributed to RNAi, but studies have suggested that some viruses can exclude highly similar or nearly identical viruses from the same cells by using a protein-based mechanism that acts on virus replication [8,9]. To complicate matters, many studies have now shown that the order of virus infection could change how viruses interact. The sequential inoculation of potato virus X (PVX, a potexvirus) and potato virus Y (PVY, a potyvirus) yielded severe symptoms only when PVX was inoculated before PVY, while if PVY was inoculated before PVX symptoms were less severe and PVX titer was not increased as dramatically [1]. In a follow-up study, Goodman and Ross (1974) found that the two viruses needed to be replicating in the same cells simultaneously to obtain the maximum benefit from their interaction [10]. In another example, Chávez-Calvillo and colleagues found antagonism, when papaya ringspot virus (PRSV, genus Potyvirus) infection occurred after papaya mosaic virus (PapMV, genus Potexvirus) inoculation in papaya, and synergism when PRSV was inoculated 30 days before or together with PapMV [4]. Understanding how viruses interact within a shared host is crucial for developing effective disease management strategies, particularly for economically important pathogens.

Tomato spotted wilt virus (TSWV) and impatiens necrotic spot virus (INSV) are two closely related viruses within the Orthotospovirus genus that share hundreds of plant hosts and a dozen vectors, primarily *Frankliniella* thrips species [11,12]. Both viruses have tripartite negative-sense RNA genomes that encode for an RNA-dependent RNA polymerase (large or L segment), for polyglycoproteins and movement protein (medium or M segment), and for nucleocapsid proteins and a silencing suppressor (small or S segment). The TSWV and INSV isolates we used in this study [13,14] share less than 80% sequence similarity in the S region, coding for the nucleocapsid protein, the sequence most used to define orthotospoviruses at the species level. Mixed infections of TSWV and INSV were first reported in tomato plants from Italy in 2000 [15]. In the US, they were reported in tobacco in six counties in Georgia, Florida, South Carolina, and Virginia in 2002 [16], where the incidence of co-infection in tobacco was around 40% [17], and in North Carolina and Kentucky since 2003 [16]. Because INSV single infection in the study by Martinez-Ochoa was never detected in tobacco but only in the surrounding weeds, authors suggested that INSV could act as a helper virus during mixed infection, suggesting that TSWV and INSV have a synergistic relationship during mixed infection [17]. On the contrary, in our studies that used two Pennsylvania isolates of the two viruses, INSV and TSWV showed an antagonistic relationship in their insect host, where INSV negatively impacted the ability of thrips to retain and transmit TSWV and suggesting that vectors could serve as bottleneck for the establishment and maintenance of TSWV and INSV mixed infection [18]. While our study did not find any significant difference between the identity of plant volatiles emitted during single and double infection, or between symptoms and infected plant appearance and size, thrips were able to distinguish between treatments, suggesting that plants respond differently when infected by one or two viruses [18]. Hypothetically, the result and establishment of a mixed infection could depend on the direct interaction of the two viruses (for instance, on direct competition for host resources), on plant responses to infection, and on the propensity of the virus vectors two transmit one virus, the other of both simultaneously.

RNA interference (RNAi) is one of the primary defense mechanisms employed by plants against viral infections and is a highly conserved and sequence-specific pathway that detects and degrades foreign RNA molecules [7,19,20]. RNAi is triggered by the presence of (dsRNA) formed during viral replication or hairpin RNA produced via self-complementarity [21,22]. These RNA forms elicit their processing via plant Dicer-like (DCL) enzymes into small interfering RNAs (siRNAs) of approximately 21–24 nt [21,22,23]. siRNAs are then loaded into Argonaute (AGO) proteins within the RNA-induced silencing complex (RISC) to guide the degradation of complementary viral RNA [22,24]. Virus-derived small RNAs (vsRNAs) accumulate during infection and serve as markers of RNAi activity, reflecting the host’s antiviral defense status. Additionally, plant microRNAs (miRNAs) regulate endogenous gene expression and can be influenced by viral infection, stress, and immune responses [25,26,27,28]. In mixed infections, competition for the host RNAi machinery may influence viral accumulation and host responses, potentially determining which virus dominates [29,30]. Orthotospoviruses have evolved counter-defense strategies to suppress RNAi [19]. The non-structural NSs protein, a well-characterized RNAi suppressor, inhibits the RNA silencing machinery by binding siRNAs and preventing their incorporation into RISC [31,32,33]. This suppression is crucial for viral replication and movement within the host. In a mixed infection scenario, the balance between viral RNAi suppressors and host antiviral responses could shift, potentially leading to the dominance of one virus over the other [34,35]. However, the extent to which RNAi influences viral interactions and whether one virus can outcompete the other in plant hosts remains unknown.

This study aims to elucidate the molecular interactions between TSWV and INSV in *Nicotiana benthamiana* by examining viral accumulation, vsRNA profiles, and miRNA regulation in single and mixed infections. By integrating ELISA-based viral quantification with high-throughput small RNA sequencing, we seek to determine whether RNAi-mediated mechanisms contribute to viral suppression and to identify host miRNA responses that may influence viral competition. Specifically, we aim to (1) assess whether TSWV and INSV exhibit competitive or cooperative interactions in plant hosts, (2) analyze the distribution and abundance of vsRNAs in single and mixed infections, and (3) identify differentially regulated host miRNAs that may mediate virus–virus interactions. Understanding these interactions will provide insights into the broader role of RNAi in shaping plant–virus relationships and inform strategies for managing orthotospovirus diseases in agriculture.

## 2. Materials and Methods

### 2.1. Viruses and Their Maintenance

*N. benthamiana* plants at the 3–4-leaf stage and grown from seed were mechanically inoculated with TSWV isolate PA01 [13] and/or INSV isolate UP01 [14], using virus-infected tissue from the originally infected hosts stored at −80 °C as a source of inoculum, as in [13,14]. Inoculated plants were maintained in a growth chamber at a temperature of 25 °C with a 16/8 h light/dark photoperiod.

### 2.2. Virus–Virus Interaction

Since infectious clones of these viruses were not available at the time of the experiment, virus–virus interaction assays were conducted using the mechanical inoculation of infected plant extract. For single infections, *N. benthamiana* plants were inoculated with five different dilutions of a single-virus inoculum (X, 0.8X, 0.6X, 0.4X, and 0.2X), which were prepared by adding the sap of healthy plants as described in Figure 1. In a separate group, plants were inoculated with the same dilutions of one virus (X, 0.8X, 0.6X, 0.4X, and 0.2X), but each dilution was mixed with an equal volume of inoculum containing the second virus, creating a mixed infection (Figure 1). For the single-virus inoculum, healthy sap was substituted with the inoculum containing the second virus. For each dilution, five plants were inoculated, giving a total of 25 plants per treatment (single or mixed infection). The same number of plants was inoculated in a reciprocal manner, with the dilutions of the second virus mixed with the same volume of the inoculum with the first virus. Systemic leaves were checked for symptom appearance two weeks after inoculation and tested with ELISA, as described in Section 2.4. The results from both groups were then compared to assess the effects of co-infection. The experiment was repeated twice to ensure reproducibility.

### 2.3. Sequential Inoculation for Study of Super Infection Exclusion

*N. benthamiana* plants were mechanically inoculated with the two viruses sequentially at a one-hour interval, each on two different leaves and overlapping on a third leaf, for a total of five leaves per plant (described in Figure 2). Plants were inoculated with five different dilutions of one virus inoculum (X, 0.8X, 0.6X, 0.4X, and 0.2X) obtained with the addition of sap from a healthy plant, with five plants per dilution, resulting in a total of 25 plants per combination. An equal volume of inoculum containing the second virus was inoculated as explained above after one hour. Specifically, 25 plants were inoculated with TSWV diluted and spiked with INSV, and 25 plants with INSV diluted and spiked with TSWV. As controls, three leaves per plant were inoculated with a single virus at the same inoculum dilutions without any additional inoculation (25 plants with TSWV diluted and 25 plants with INSV diluted). The experiment was repeated twice. Systemic leaves were checked for symptom development and tested by ELISA, as described in Section 2.4.

For another experiment, INSV or TSWV inoculum was inoculated on the left half of two leaves in each plant. Forty-eight h after the first inoculation, the right half of the same two leaves was inoculated with equal amount (*w*/*w*) of the other virus. Systemic leaves were tested with ELISA as in Section 2.4. The experiment was repeated twice.

### 2.4. ELISA Testing

Three leaf disks (one per leaf) were collected from systemically infected leaves two weeks post-inoculation. For asymptomatic plants, an equal number of leaf disks were sampled from systemic leaves. These samples were used for virus titer quantification via ELISA (Agdia, Elkhart, IN, USA), following the manufacturer’s protocol specific to each virus. As positive and negative controls, leaf samples from previously confirmed infected plants and healthy (uninoculated) plants were included, respectively. ELISA was performed in 96-well high-bind–solid-microtiter plates (Agdia, Elkhart, IN, USA), and absorbance was measured at O.D._405nm_ using a Dynex Opsys MR™ microplate reader (Aspect Scientific Ltd., Cheshire, UK). Negative controls had OD values of approximately 0.2, and samples with O.D. readings exceeding three times that of the negative controls were considered positive. Most samples, including the positive controls, had O.D. ranging from 1.5 to 2.0. ELISA O.D. values were used as a representation of viral titer.

### 2.5. Statistical Analysis

The ELISA results were used to perform linear regression analyses using O.D. values from each virus separately to determine whether viral titers differed between single and mixed infections across inoculum dilutions. The regression analyses compared the trend generated by 25 plants per treatment. To assess whether the overall trend of the regression line (i.e., the pattern of response) varied between single and mixed infections, we used a multivariate analysis of covariance (MANCOVA).

A MANCOVA was chosen because it allows for the simultaneous analysis of multiple dependent variables. In this case, the independent variable (inoculum dilutions) was tested for its effect on two dependent variables (O.D. values from single and mixed infections). To evaluate the effect, we used *Pillai’s trace*, a multivariate test statistic provided via MANCOVA that measures how much variance in the dependent variables is explained by the independent variable. Higher values of *Pillai’s trace* indicate a stronger difference between groups, with the following general interpretation—≥0.8: strong effect; 0.4–0.7: moderate effect; and <0.4: weak effect. However, while a higher Pillai’s trace value suggests higher separation between groups, it may not be statistically significant if the *p*-value is above the significance threshold (*p* < 0.05 in our case).

To evaluate whether our sample size provided sufficient statistical power, we performed a post hoc power analysis using the observed Pillai’s trace value (V) and the R package “PWR”. The effect size was converted to Cohen’s f^2^ using the formula f^2^ = V/(1 − V) and used in a multiple regression power calculation at a significant level of 0.05. The resulting statistical power was calculated to be >0.99, indicating that the analysis had sufficient power to detect the observed multivariate effect with high confidence.

### 2.6. Virus Inoculations for sRNA Sequencing

A batch of 10 *N. benthamiana* plants for each single infection or mock inoculation and 20 plants for mixed-infection were mechanically inoculated using the following scheme—Treatment 1: TSWV PA01 and healthy *N. benthamiana* (1:1, *w*/*w*); Treatment 2: INSV UP01 and healthy *N. benthamiana* (1:1, *w*/*w*); Treatment 3: mixture of TSWV PA01 and INSV UP01 (5:2, *w*/*w*) as the source of mixed-infection; and Treatment 4: mock-inoculated with healthy *N. benthamiana* leaves [13,14].

### 2.7. RNA Extraction, sRNA Sequencing, and qPCR

Three systemically infected leaves from each plant were sampled and stored at −80 °C. Three leaf disks were sampled from those three systemically infected leaves (one leaf disk per leaf) for virus titer quantification by ELISA. Leaf tissue showing no statistically significant differences in virus titer at the intra-treatment level (in the same treatment) was used for RNA extraction and library preparation. Three leaf disks (one leaf disk per leaf) of two plants in the same treatment group were pooled (6 leaf disks from 2 plants) to generate one library for sRNA sequencing. Two libraries were performed for each treatment to generate biological replicates. Total RNA was extracted with the Quick-RNA miniprep kit (Zymoresearch, Irvine, CA, USA) following the manufacturer protocol. RNA was quantified using a NanoDrop 2000 spectrophotometer (Thermo Scientific, Waltham, MA, USA).

Aliquots containing 500 ng of total RNA were sent to the Genomics Core Facility at the Pennsylvania State University for sequencing. RNA quantity was measured using a Qubit (Thermo Fisher, Waltham, MA, USA) and quality was measured using an Agilent Bioanalyzer (Santa Clara, CA, USA). Libraries were assembled using the TruSeq Small RNA Library Preparation Kit following the manufacture protocol (Illumina, San Diego, CA, USA) and then sequenced with an Illumina NextSeq 550.

qPCRs on the RNA sent for sequencing were performed using primers, TSWV-L-For: TCTCCACCTCGCTTCTTTGT and TSWV-L-Rev: AAACAAAGGGATGGCAACTG for TSWV [13], and INSV-L-For: AGAGAGGACCACCCTTGGAT and INSV-L-Rev: ATGTTCGGTGAGCTGGTTTC for INSV [14]. Products were amplified using iTaq Universal SYBR Green Supermix following the manufacture protocol (Bio-Rad, Hercules, CA, USA).

### 2.8. Bioinformatic Analysis

Small RNA reads were trimmed for adapter sequences and mapped against the viral (NCBI accession numbers KT160280-KT160282 for TSWV and MH171172–MH171174 for INSV) [13,14] and host genomes (*N. benthamiana* genome version 0.5) [36] using Shortstack version 3.8.5 [37], allowing zero mismatches. Read size, distribution along the genomic segments and along single ORFs, polarity, 5′-nt enrichment, and hotspot analyses were performed using SAMtools version 1.9 [38] and in-house Perl scripts, and imaged using Microsoft Excel^®^ v. 10. Relative frequencies of 5′ terminal nucleotides and vsRNAs hotspot profiles were generated by MISIS 2.7 [39]. A differential expression of miRNA loci was analyzed using DESeq2 with a log_2_ fold threshold of 1 and an alpha of 0.1. The Benjamini–Hochberg procedure was used for multiple testing of *p* values’ adjustment.

## 3. Results and Discussion

### 3.1. INSV and TSWV Have an Antagonistic Interaction

Our data showed that the number of infected plants in mixed infections differed from those in single infections, despite using the same inoculum, at the same dilution and ratio (Table 1) and performing the experiment at the same time. This indicates that there is an interaction between the two viruses. Disease incidence (DI) confirmed with ELISA, showed that, in single infections, INSV infected 20–60% of the plants across the dilutions in experiment 1 (Table 1) and 60–100% in experiment 2 (Appendix A). In mixed infections, however, INSV was detected in 0–60% of plants in experiment 1, with three of five dilutions showing no infection (0% DI), while in experiment 2, DI ranged from 0 to 100%. In the case of TSWV, 60–100% of the plants in single infections were found to be positive, whereas mixed infections of TSWV showed 0–60% DI in experiment 1 (Table 1). In experiment 2, DI ranged from 0 to 100% in single infections and 0 to 20% in mixed infections. This variability in disease incidence may be due to inconsistencies in the initial virus amount in the inoculum. However, both experiments consistently showed that single infections resulted in higher DI than mixed infections. Interestingly, in mixed infections, the second virus was detected in all plants (100% DI) (Table 1 and Appendix A), regardless of the detection of the first virus, except for TSWV in experiment 2, which showed that 20–60% plants were positive across the dilutions (Appendix A). These results suggest that the presence of one virus negatively affects the infection success of the other. However, neither virus appears to offer a fitness advantage over the other in mixed infections. ELISA and subsequent analysis using the O.D. showed a clear difference in the pattern of regression lines for single and mixed infections for both viruses, as evidenced by the significant *p*-values (INSV: *p* = 0.03, TWSV: *p* = 0.001) (Figure 1).

### 3.2. Host Plant Appears to Contribute to the Antagonism

In the previous experiment, we found that mixing two viruses in the inoculum caused an antagonistic interaction in the host plant. However, this result did not clarify whether the antagonism was due to the interaction between the viruses or if the host plant also contributed. To address this question, we designed two sets of experiments.

In the first experiment, we inoculated the viruses sequentially on different leaves, with a one-hour interval between inoculations (Section 2.3). ELISA, followed by statistical analysis on the regression lines showed higher *Pillai’s Trace* values (INSV: 0.78; TSWV: 0.79) in both viruses, indicating a strong effect (details on *Pillai’s trace* in Section 2.5). However, the associated *p*-values were not significant (INSV: *p* = 0.22; TSWV: *p* = 0.21; Figure 2). These results indicate that the antagonism between the viruses was less pronounced in this experiment compared to our previous one. However, the viral titers were always lower in the mixed infection as before, suggesting that the host plant contributes to this antagonism, since the viruses were inoculated on different leaves and with a time interval.

To verify the plant’s contribution and explore the potential super-infection exclusion or cross-protection, we conducted a second experiment in which INSV and TSWV were inoculated sequentially with a 48 h interval (Section 2.3). This design allowed us to test whether the first virus inoculated influenced the infection success of the second. We detected mixed infections in several plants under these conditions. More specifically, when INSV was inoculated first, the mixed infection rate was 29.4% (n = 34; 10 plants were infected with both viruses, 21 with only INSV, and 3 with only TSWV). When TSWV was inoculated first, the mixed infection rate was 43.3% (n = 30; 13 plants were infected with both viruses, 12 with only INSV, and 5 with only TSWV). We also concomitantly inoculated plants with a single virus as control, which had average infection rates of 96.7% for INSV and 93.3% for TSWV (n = 30). This result suggests no evidence of super-infection exclusion between INSV and TSWV. Both viruses were capable of establishing infections, regardless of the inoculation order. Nonetheless, the number of plants that got infected in mixed infections was less than the ones in single infections. This again reinforces the host’s contribution in the antagonism.

### 3.3. Small RNA (sRNA) Profiles Reflect Viral Antagonism

Since our ELISA data suggested a role of the host plant in the dynamics of mixed infection, we wanted to look at the antagonism and host’s contribution at the molecular level. To do so, we conducted sRNAs profiling using plants that were inoculated with the ratio that gave us a consistent number of successful mixed infection (TSWV: INSV = 5:2; *w*/*w*). Before sequencing, only plants with O.D. in ELISA tests that were similar for INSV and TSWV in all treatments (single TSWV, single INSV, as well as mixed infections) were selected for sRNAs sequencing. Two biological replicates were used per treatment, each generated by pooling three leaf disks each from two infected plants, thus a total of four plants per treatment. The ELISA results are based on the amount of capsid protein and thus are a good proxy for virions. To complement this, we also quantified viral RNA levels by qPCR targeting the RdRp genes of both viruses. Interestingly, while ELISA readings were similar for both viruses, qPCR found lower Ct values (i.e., higher RNA levels) for INSV compared to TSWV. The INSV:TSWV ratios of RdRp abundance were 1:0.56 and 1:0.34 for replicates 1 and 2, respectively, indicating lower TSWV RNA levels. This discrepancy between ELISA and qPCR results suggests that protein and RNA quantification may reflect different aspects of viral accumulation and might be due to the differences in the quantity of virions vs. RNA, or in different efficiency of ELISA tests that were performed using commercial kits.

After sequencing, reads that were mapped to either viral or host genome with zero mismatches were counted for downstream analysis. We chose zero mismatches to minimize false positives alignments and to ensure high-confidence mapping, particularly given the poor genome annotation for *N. benthamiana*. While this conservative approach strengthens the reliability of our mapping results, it may also exclude sRNA variants that contain sequencing errors, post-transcriptional modifications, or natural sequence polymorphisms. As such, our results may underestimate the full diversity of virus-derived sRNAs, particularly in regions of the genome prone to variation.

After mapping, we found, in mock-inoculated plants, that 60.8% and 62.5% of sRNAs were of host origin (Figure 3). The remaining sRNAs not mapped to the host genome likely represent unannotated plant sequences, endophytic contaminants, or degradation products that do not align to the genome under our mapping criteria. In INSV-infected plants, host-derived sRNAs decreased to 34.0% and 36.8%, while 34.0% and 29.4% mapped to the INSV genome (Figure 3). A similar trend was observed in TSWV-infected samples, but with a higher proportion of viral small RNAs (vsRNAs) (47.9% and 43.0%) and lower host sRNAs (26.6% and 30.5%) (Figure 3).

In mixed infections, vsRNAs from both TSWV and INSV were detected. However, TSWV-derived vsRNAs were significantly less abundant (4.1% and 4.2%) compared to TSWV single infections (47.9% and 43%) and INSV-derived vsRNAs in mixed infections (31.8% and 27.1%) (Figure 3). This trend aligned with qPCR results for RdRp genes and suggests a biological antagonism. While the exact mechanism is not clear, it could be due to the antagonistic interaction between INSV and TSWV in mixed infection, or an elevated antagonistic response from the host, especially toward TSWV. Host sRNAs levels in mixed infections (33.7% and 34.6%) were comparable to those in single infections.

We then mapped and visualized vsRNAs across the viral genomes, revealing distinct distribution patterns. INSV vsRNA hotspots were similarly distributed on both positive and negative strands in single and mixed infections (Figure 4a). However, TSWV vsRNA distribution shifted dramatically from single to mixed infection, decreasing at the S segment (position 13,689–16,663) and reaching a high amount at the M segment (position 8915–13,679) (Figure 4b). This TSWV-specific change suggests a virus–virus interaction, a difference in the secondary structure of the RNA and of its accessibility to the RNAi machinery, and/or a unique host-virus dynamic. Consistent vsRNA profiles across biological replicates were found for single and mixed infections of both viruses (Appendix A).

### 3.4. Different DCLs and AGOs Could Be Recruited During Co-Infection

sRNAs of 24nt in length were abundant in mock-inoculated plants (Figure 5a), whereas most of the host endogenous sRNA (Figure 5a) and vsRNA in infected samples were 21nt and 22nt (Figure 5b,c). For INSV in single, as well as mixed, infection, 21nt and 22nt vsRNAs had the highest abundance (Figure 5b and Appendix A), while for TSWV in single infection, 22nt vsRNAs had the highest level of accumulation (Figure 5c) for both biological replicates (Appendix A). However, vsRNAs of TSWV in mixed infection varied from the single infection and the % of 21 and 22nt were similar to each other (Figure 5c). These trends were maintained across biological replicates (Appendix A). A high level of accumulation of 21nt and 22nt vsRNAs has been seen in other virus-infected plants, and different DCLs were involved during vsRNAs generation [40,41]. In *A. thaliana*, DCL4 is predominantly responsible for the production of 21nt vsRNAs, and DCL2 for the production of 22nt vsRNAs [40,41]. The highest abundance of 21nt vsRNA was in fact observed with a TSWV American isolate [42] and an Italian wild-type isolate: p202/3WT [43]. On the contrary, we found that 22nt had the highest level of accumulation in TSWV, which was also reported for a second Italian TSWV isolate, together with the above-mentioned p202/3WT [43]. This suggests that DCL2 might play a crucial role during TSWV infection, but not in TSWV and INSV mixed infection. While the previous studies indicate that the use of DCLs was isolate-dependent, our study infers that it was species-specific. Our results suggest that DCL4 and DCL2 play a major role during infections, and that INSV and TSWV and their combination elicit DCLs differently.

### 3.5. Virus-Activated Small Interfering RNAs (vasiRNAs) in Virus-Infected Plants

During a virus infection, virus-activated small interfering RNAs (vasiRNAs) are synthesized from host-gene transcripts and differ from the other endogenous sRNAs of host origin. VasiRNAs, mostly 21 and 22nt in length incorporated into AGO1 and AGO2/RISC, cause the silencing of target mRNAs of host genes [44]. vasiRNAs were first described in 2014 [44], when *Arabidopsis thaliana* infected with silencing suppressor-deficient viruses accumulated endogenous sRNAs predominantly 21nt in length. In our experiment, both mock-inoculated controls showed the highest abundance of 24nt sRNAs, which is the typical size of host endogenous sRNAs, and a lower abundance of 21, 22, and 23nt sRNAs (Figure 5a). In contrast, most host-endogenous sRNA in infected samples were 21nt or 22nt in length (Figure 5a), which indicates the possibility of vasiRNAs production in the infected plants. In *A. thaliana* DL4/RDR1, AVI2H, and AGO2 are required for vasiRNA biogenesis [44]; however, nothing is known about this same process in *N. benthamiana*, and mutants for these enzymes should be used to characterize this process in this host.

### 3.6. Conserved Host Response in miRNAs Regulation

We analyzed miRNA profiles in single and mixed infections to determine whether the plant exhibits distinct responses to co-infection. Principal component analysis (PCA) revealed distinct clustering, with mock-inoculated and mixed-infected treatments grouping separately from single infections (Figure 6a). Among 174 identified *miRNAs*, 80 (20 known miRNA families and 60 novel miRNAs) were significantly differentially expressed (based on FDR ≤ 0.1, FoldChange > 2) in at least one comparison (Figure 6, Appendix A). All 51 *miRNAs* differentially regulated in mixed infections were also differentially expressed in TSWV and/or INSV single infections, following the same trend (up- or downregulated). This suggests that mixed infections do not induce unique miRNA responses, compared to single infections. Among the known miRNAs, the miR398 family was significantly upregulated in all comparisons (Figure 6b–d). miR398, a key regulator of oxidative stress [45] has been shown to be upregulated in tomato plants during infections by both persistent (southern tomato virus) and acute plant viruses (rice stripe virus) [46,47]. miR398 primarily targets antioxidant-related genes, especially Cu/Zn superoxide dismutases (CSD1 and CSD2), which reduce reactive oxygen species (ROS) and protect cells from oxidative damage [45,48]. In a recent study, Lin et al. found that bamboo mosaic virus (BaMV) enhances viral accumulation and symptom severity in *N. benthamiana* by upregulating miR398, followed by suppressing CSD2 and increasing ROS [49]. Similarly, in maize, elevated levels of miR398b resulted in a reduced expression of CSD genes, increased ROS, and enhanced susceptibility to sugarcane mosaic virus (SCMV) [50]. These findings suggest that viruses may facilitate infection and symptom development by exploiting the miR398-mediated regulation of antioxidant genes. In addition to miR398, we have found that *miR170* and *miR395* were highly upregulated in TSWV-infected plants (Appendix A). miR170 is involved in plant immune signaling, while miR395 regulates sulfate assimilation, and both have been previously associated with plant–virus interactions [51,52]. The similarity in miRNA regulation across single and mixed infections suggests that orthotospovirus infections activate conserved host responses, rather than inducing novel miRNA pathways.

## 4. Conclusions

This study demonstrates that INSV and TSWV have an antagonistic relationship in *N. benthamiana*, which is a well-established model for plant–virus interaction studies and has consistently provided reliable disease incidence and severity in our prior work. Its large leaf size and susceptibility also offered practical advantages for mechanical inoculation, particularly when applying overlapping inoculum for co-infection experiments. However, *N. benthamiana* is not a natural host for either TSWV or INSV, and virus–virus and virus–host interactions can vary across plant species. Hence, while our results offer valuable molecular insights into orthotospovirus interactions in this model system, it may vary in natural hosts.

With small RNA sequencing, we found a similar level of INSV vsRNAs in single and mixed infection; however, TSWV vsRNAs were much lower in mixed infection compared to TSWV single infections and INSV vsRNAs in mixed infections. While the exact mechanism explaining this antagonistic relationship is unclear, our study provides insights into varied host responses involving RNAi to single and mixed orthotospovirus infection. Similarly, miRNA accumulation did not change much between INSV single and mixed infections, but TSWV showed higher accumulation during single infection. Our study indicates that, while plants use conserved mechanisms to act against viruses, differences must exist concerning where or how orthotospoviruses interact with the plant RNAi machinery. In particular, the observed shifts in vsRNA size profiles point toward differential recruitment of Dicer-like (DCL) enzymes, such as DCL2 and DCL4, during co-infection. Although we did not directly assess the expression of RNA silencing components like DCLs or AGOs in this study, our findings highlight this as an important direction for future work. Further investigation of these components may help elucidate the molecular basis of antagonism and host-mediated suppression.

Altogether, this work provides new insights into how orthotospoviruses interact within a shared host and lays the groundwork for developing species- or genus-specific management strategies that exploit these interactions for more effective disease control. 

## Figures and Tables

**Figure 1 viruses-17-00789-f001:**
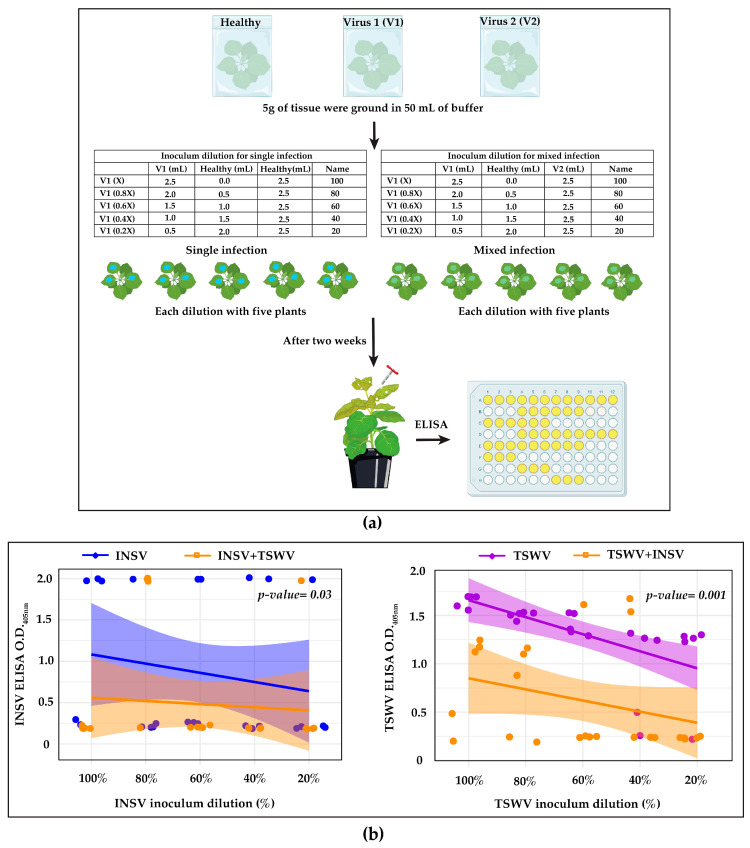
Sample preparation for ELISA (**a**) and data showing antagonistic interaction of INSV and TSWV (**b**). The X-axis represents the percentage of virus inoculum, as shown in (**a**), and the Y-axis indicates the O.D._405nm_ values obtained from ELISA. **Left panel** (**b**): INSV titer was reduced in mixed infections (orange line) compared to single INSV infections (blue line). **Right panel** (**b**): similarly, the TSWV titer (orange line) in mixed infection was lower than in single TSWV infections (violet line). *N. benthamiana* plants were mechanically inoculated using five different dilutions (five plants per dilution) for each virus. ELISA results from single-virus infections were compared with those from mixed infections, where each dilution was spiked with an equal amount of the second virus. Statistical analysis using MANCOVA showed a significant decrease in viral titer for both viruses in mixed infections compared to single infections, with *p*-values confirming this effect across inoculum dilutions.

**Figure 2 viruses-17-00789-f002:**
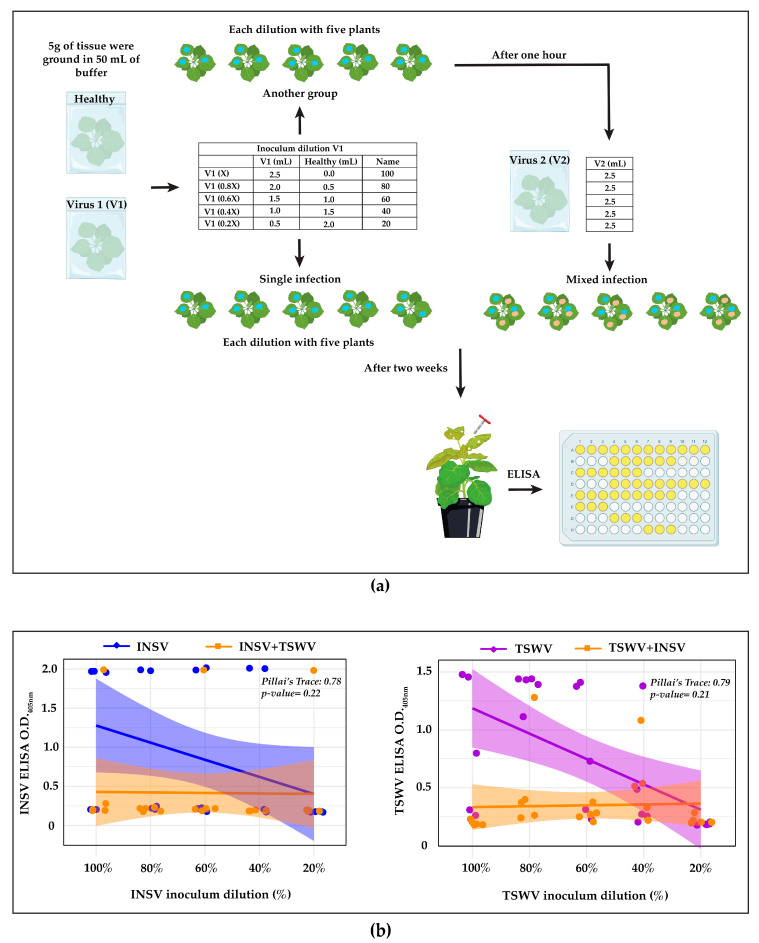
Sample preparation for ELISA (**a**), and data showing antagonistic interaction of INSV and TSWV when they were inoculated sequentially on different leaves, with a one-hour interval between inoculations (**b**). The X-axis represents the percentage of virus inoculum as shown in (**a**), and the Y-axis indicates the O.D._405nm_ values obtained from ELISA. **Left panel** (**b**): INSV titer was reduced in mixed infections (orange line) compared to single INSV infections (blue line). **Right panel** (**b**): Similarly, TSWV titer (orange line) in mixed infection was lower than in single TSWV infections (violet line). We used five different inoculum dilutions (X, 0.8X, 0.6X, 0.4X, and 0.2X), with five plants per dilution for each virus and analyzed the differences between the regression lines of single and mixed infections using MANCOVA. Although the O.Ds. were lower in the mixed infections than the single infections in both cases, it was not significant (*p*-values were higher than 0.05). However, the higher Pillai’s trace values (INSV: 0.78; TSWV: 0.79) indicate a strong group effect.

**Figure 3 viruses-17-00789-f003:**
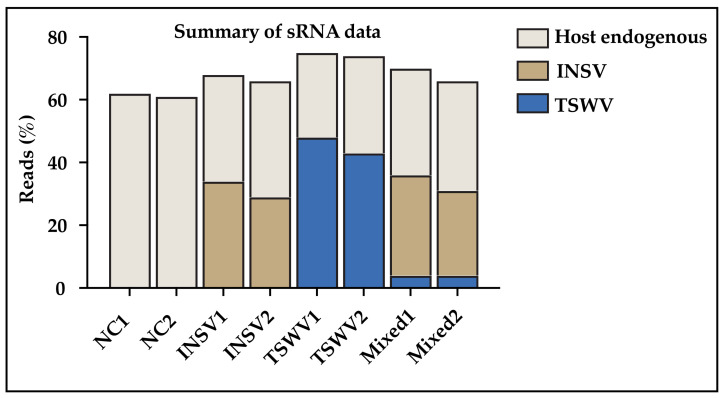
Summary of small RNA reads.

**Figure 4 viruses-17-00789-f004:**
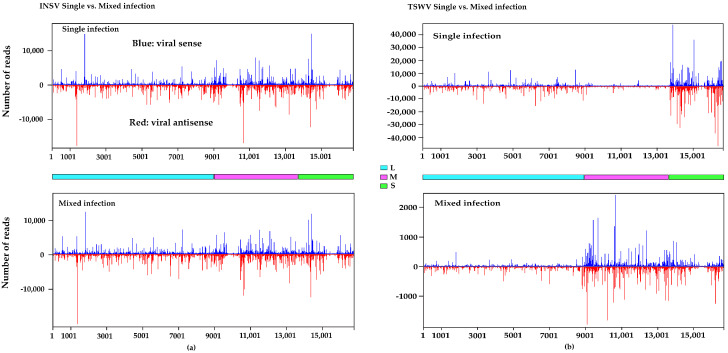
Hotspot distributions on the viral genome. Peaks represent multiple reads aligned to the genome in the same position. Y-axis: number of reads. X-axis: nucleotide position on the viral genome. (**a**) INSV single and mixed infection. (**b**) TSWV single and mixed infection. Blue: viral sense; (Red: viral antisense. Figures were generated using MISIS. L, M, and S segments are represented by the consecutive segments with color codes on the solid lines between the top and bottom panels. For INSV, L segment: 1–8774nt, M segment: 8775–13,751nt, and S segment: 13,752–16,761nt. For TSWV, L segment: 1–8914nt, M segment: 8915–13,679nt, and S segment: 13,689–16,663nt.

**Figure 5 viruses-17-00789-f005:**
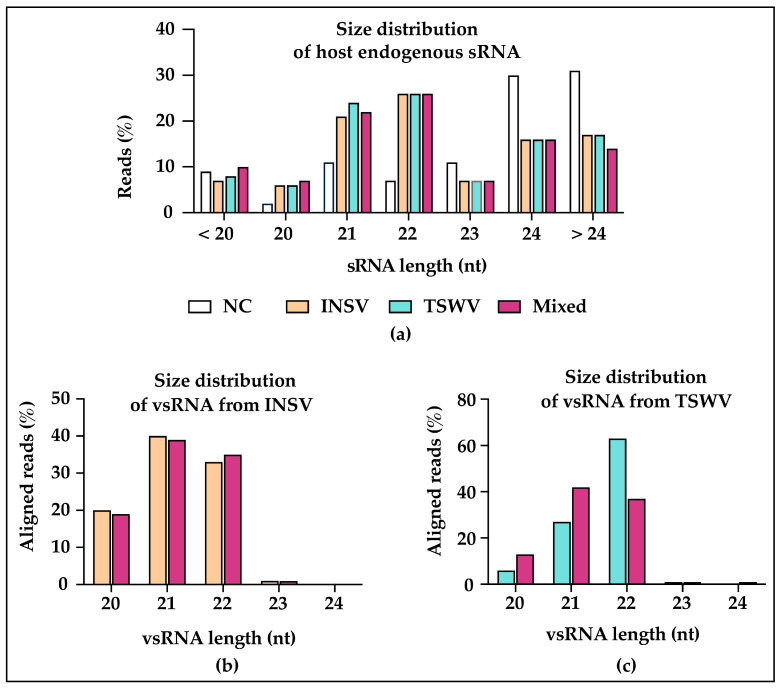
Summary of small RNA reads based on their size. NC: mock-inoculated negative control. Size distribution of host-endogenous sRNAs, vsRNAs from INSV, and vsRNAs from TSWV (**a**–**c**).

**Figure 6 viruses-17-00789-f006:**
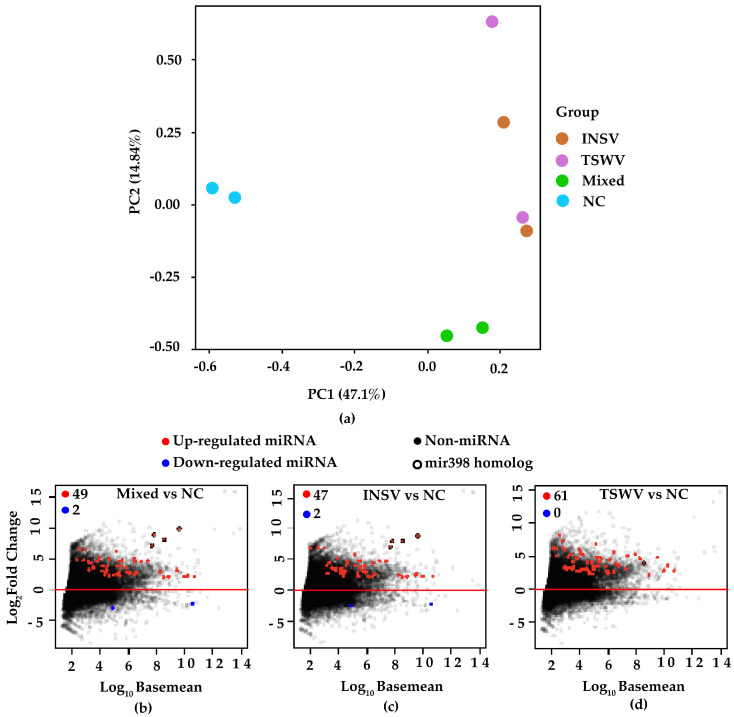
Host miRNAs and their differential regulation in different treatments (NC: mock-inoculated control, INSV, TSWV, and mixed infections. (**a**) PCA plot based on de novo identified host miRNA clusters. (**b**–**d**) Mean abundance plot of *N. benthamiana* miRNA loci comparing two different treatments. Significantly upregulated and downregulated miRNA loci are highlighted (alternative hypothesis: FDR ≤ 0.1, true difference > 2-fold after Benjamini–Hochberg correction for multiple testing).

**Table 1 viruses-17-00789-t001:** Virus–virus interaction experiment setup and percentage of disease incidence confirmed by ELISA.

INSV	Disease	Disease Incidence			
**Single**	**Incidence**	**INSV Mixed**	**INSV Infected**	**TSWV Infected**	**Mixed-Infected**
INSV (X)	60%	INSV (X) + TSWV (X)	0%	100%	0%
INSV (0.8X)	20%	INSV (0.8X) + TSWV (X)	60%	100%	60%
INSV (0.6X)	40%	INSV (0.6X) + TSWV (X)	0%	100%	0%
INSV (0.4X)	40%	INSV (0.4X) + TSWV (X)	0%	100%	0%
INSV (0.2X)	20%	INSV (0.2X) + TSWV (X)	20%	100%	20%
**TSWV**	**Disease**	**Disease Incidence**			
**Single**	**Incidence**	**TSWV Mixed**	**TSWV Infected**	**INSV Infected**	**Mixed-Infected**
TSWV (X)	100%	TSWV (X) + INSV (X)	60%	100%	60%
TSWV (0.8X)	100%	TSWV (0.8X) + INSV (X)	60%	100%	60%
TSWV (0.6X)	100%	TSWV (0.6X) + INSV (X)	20%	100%	20%
TSWV (0.4X)	60%	TSWV (0.4X) + INSV (X)	40%	100%	40%
TSWV (0.2X)	80%	TSWV (0.2X) + INSV (X)	0%	100%	0%

## Data Availability

The datasets generated and analyzed for this study can be found in the NCBI with Accession Number: PRJNA833054.

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
