# Peer review of "Antagonism in Orthotospoviruses Is Reflected in Plant Small RNA Profile"

_viruses, 2025, doi:10.3390/v17060789_

Round 1
Reviewer 1 Report (New Reviewer)
Comments and Suggestions for Authors
This study experimentally revealed the antagonistic effects of tomato spot wilt virus (TSWV) and Impatiens necrotic spot virus (INSV) in plant hosts, and combined with small RNA sequencing, analyzed the potential regulatory role of the host RNAi mechanism. The topic selection of the thesis is of great scientific significance. The overall experimental design is reasonable, the data analysis method is appropriate, and the conclusion contributes to understanding the virus interaction and host defense mechanism. However, some experimental details and result explanations need to be further clarified or supplemented.
- In the experiment, only five plants were used at each dilution, and the infection rates of some treatment groups (such as mixed infections) fluctuated greatly (for example, INSV was 0-60% in mixed infections), which might affect the statistical power. It is suggested to supplement the sample size or provide statistical power analysis to support the robustness of the conclusion.
- ELISA showed that the viral titers of TSWV and INSV were similar in the mixed infection, but qPCR showed that the RNA amount of TSWV was significantly lower than that of INSV. This contradiction requires further explanation. For example: Is it due to the difference in the assembly efficiency of viral particles that the amount of capsid protein (CP) detected by ELISA is inconsistent with that of RdRp RNA detected by qPCR? Is the sensitivity of ELISA detection affected by cross-reactions or background signals? It is recommended to supplement Western blot or electron microscope data to verify the abundance of virus particles.
- ELISA showed that the viral titers of TSWV and INSV were similar in the mixed infection, but qPCR showed that the RNA amount of TSWV was significantly lower than that of INSV. This contradiction requires further explanation. For example: Is it due to the difference in the assembly efficiency of viral particles that the amount of capsid protein (CP) detected by ELISA is inconsistent with that of RdRp RNA detected by qPCR? Is the sensitivity of ELISA detection affected by cross-reactions or background signals? It is recommended to supplement Western blot or electron microscope data to verify the abundance of virus particles.
Biological duplication (only twice) may not be sufficient to cover biological variations, and this limitation needs to be clarified in the discussion.
- The experiment only used Nicotiana benthamiana, while the natural host range of TSWV and INSV was extensive. The host specificity of the results should be emphasized in the discussion to avoid overgeneralization to other plant systems.
- The annotations in Figures 1 and 2 need to be clearer. For instance, the horizontal and vertical coordinate units and significance markers (such as P-values) should be clearly marked.
Figures S1-S3 need to provide more detailed legends to illustrate the specific differences among different treatments.
- The "dilution X" for virus inoculation needs to clearly specify the exact concentration (such as the virus content per unit volume) to enhance the repeatability of the experiment.
- The alignment criterion of "zero mismatch" in small RNA sequencing may miss some variant sequences. It is necessary to explain the basis for choosing this threshold.
- In discussion, the mechanisms of viral interactions in previous studies (such as protein-mediated rejection vs. RNAi competition) can be compared to further explain the dominant role of host RNAi in this study. When referring to the conserved regulation of mirRNA such as miR398, it is recommended to link the functions of their downstream target genes (such as antioxidant-related genes) to enhance the mechanism explanation.
- Some of the references are rather outdated (such as the DCL-related research in 2006). It is suggested to supplement the latest progress on viral antagonism and RNAi mechanisms in recent years (such as the literature after 2020).
Author Response
This study experimentally revealed the antagonistic effects of tomato spot wilt virus (TSWV) and Impatiens necrotic spot virus (INSV) in plant hosts, and combined with small RNA sequencing, analyzed the potential regulatory role of the host RNAi mechanism. The topic selection of the thesis is of great scientific significance. The overall experimental design is reasonable, the data analysis method is appropriate, and the conclusion contributes to understanding the virus interaction and host defense mechanism. However, some experimental details and result explanations need to be further clarified or supplemented.
- In the experiment, only five plants were used at each dilution, and the infection rates of some treatment groups (such as mixed infections) fluctuated greatly (for example, INSV was 0-60% in mixed infections), which might affect the statistical power. It is suggested to supplement the sample size or provide statistical power analysis to support the robustness of the conclusion.
Response: We appreciate the reviewer’s comment regarding the number of samples at each dilution and the potential impact of fluctuation of infection rate on statistical power and we agree that 5 plants would be a problem. We performed instead linear regression analyses and Multivariate Analysis of Covariance (MANCOVA) on 25 plants per treatment, in order to avoid comparing only 5 plants per each dilution in a pairwise fashion. We added this to line 160. We rather assessed the difference between single and mixed infection across inoculum dilutions. To analyze the statistical power, we conducted a post hoc power analysis using the observed Pillai’s trace (V) from MANCOVA. The effect size was calculated using the formula (Cohen’s f² = V / [1–V]) and finally conducted the power analysis at a significance level of 0.05. We now included this description in Section 2.5, lines (174-179).
- ELISA showed that the viral titers of TSWV and INSV were similar in the mixed infection, but qPCR showed that the RNA amount of TSWV was significantly lower than that of INSV. This contradiction requires further explanation. For example: Is it due to the difference in the assembly efficiency of viral particles that the amount of capsid protein (CP) detected by ELISA is inconsistent with that of RdRp RNA detected by qPCR? Is the sensitivity of ELISA detection affected by cross-reactions or background signals? It is recommended to supplement Western blot or electron microscope data to verify the abundance of virus particles.
Response: Thank you for highlighting the discrepancy between ELISA and qPCR quantification. We appreciate reviewer’s suggestion, and we agree that this difference could be due to differences between the number of virions measured by ELISA and the amount of RNA of the L segment measure by qPCR. We acknowledge this as a limitation of our study (lines 283–297) and note it as a valuable direction for future investigation. An alternative explanation could be that the ELISA tests for INSV and TSWV have different efficiency, since they are commercial, and we also added this in our text.
- Biological duplication (only twice) may not be sufficient to cover biological variations, and this limitation needs to be clarified in the discussion.
Response: We had two biological replicates per treatment, each generated by pooling three leaf disks each from two infected plants, totaling four plants per treatment. To initially assess antagonism between the two viruses and the host response, we conducted several experiments with a larger number of samples and consistently observed similar patterns by repeating the experiments (Sections 3.1 and 3.2). Based on the reproducibility of these results and their biological plausibility, we proceeded with small RNA sequencing using two biological replicates per treatment, where each treatment reflected pooled samples from four plants to capture the representative variability. We also explicitly reported the biological variation observed between replicates to ensure transparency and avoid misinterpretation. While we appreciate this suggestion, we believe that the limited number of biological replicates did not compromise the validity of our findings.
- The experiment only used Nicotiana benthamiana, while the natural host range of TSWV and INSV was extensive. The host specificity of the results should be emphasized in the discussion to avoid overgeneralization to other plant systems.
Response: We appreciate the reviewer’s thoughtful comment. We have now discussed this in conclusion (lines 391-399), emphasizing the host-specific context of our findings and cautioning against overgeneralization.
- The annotations in Figures 1 and 2 need to be clearer. For instance, the horizontal and vertical coordinate units and significance markers (such as P-values) should be clearly marked.
Response: We have revised Figures 1 and 2 to include detailed axis labels, units, and statistical significance markers (p-values). Figure legends have also been updated accordingly for clarity.
Figures S1-S3 need to provide more detailed legends to illustrate the specific differences among different treatments.
Response: We have revised the legends for Supplementary Figures S1–S3 to clearly explain the treatment conditions, key observations, and comparisons. These updates enhance the interpretability of the supplementary data.
- The "dilution X" for virus inoculation needs to clearly specify the exact concentration (such as the virus content per unit volume) to enhance the repeatability of the experiment.
Response: Since infectious clones of these viruses are not available, we used mechanical inoculation of infected plant extract, hence the exact concentration of the viruses could not be provided. However, we have revised the figures and mentioned 5g of infected plant tissues were ground in 50mL of buffer to make the inoculum.
- The alignment criterion of "zero mismatch" in small RNA sequencing may miss some variant sequences. It is necessary to explain the basis for choosing this threshold.
Response: We agree with the reviewer that this is a conservative criterion. We chose zero-mismatch to minimize false positives and ensure high-confidence mapping. However, we have now added a rationale for this decision and included a discussion about the potential exclusion of variant sRNAs in section 3.3 (line 295-301).
- In discussion, the mechanisms of viral interactions in previous studies (such as protein-mediated rejection vs. RNAi competition) can be compared to further explain the dominant role of host RNAi in this study. When referring to the conserved regulation of mirRNA such as miR398, it is recommended to link the functions of their downstream target genes (such as antioxidant-related genes) to enhance the mechanism explanation.
Response: Thanks for this suggestion. We have discussed the mechanisms of viral interactions, such as protein-mediated rejection vs. RNAi competition in Introduction (line: 29-43). Since our study did not focus on protein-mediated cross-protection, we chose not to elaborate on it in the Results and Discussion sections.
We have now discussed about the miR398 and linked the functions of their downstream target genes (such as antioxidant-related genes) in line 373-384.
- Some of the references are rather outdated (such as the DCL-related research in 2006). It is suggested to supplement the latest progress on viral antagonism and RNAi mechanisms in recent years (such as the literature after 2020).
Response: We have added ca. 15 extra recent and relevant references throughout the manuscript.
Reviewer 2 Report (New Reviewer)
Comments and Suggestions for Authors
This is an interesting paper describing the interaction of two orthotospoviruses, TSWV and INSV, in N. benthamiana plants. The authors show that TSWV and INSV exhibit antagonistic interactions under mixed infection conditions. Furthermore, using small RNA analyses in virus-infected plants, the authors show that these interactions are controlled by the host RNA silencing machinery, which is likely to differentially process TSWV and INSV genomic RNAs in mixed infections. The results are novel and definitely worth publishing in Viruses. However, I have some concerns that should be addressed before the paper can be accepted for publication.
Main concerns
- Some data are presented in supplementary figures and tables; however, the supplementary materials have not been made available to the reviewer. Therefore, an additional round of review is required after all data, including the supplementary materials, are made available.
- In the Results and Discussion section, the discussion is provided in a minimalistic manner, so that many presented results, which are quite important, are not discussed at all. Such discussion should be included in the revised version of the paper.
Specifically, the following points should be discussed.
- a) Under mixed infection conditions, the amount of TSWV-specific sRNA was significantly lower than in TSWV-infected plants. How can this be explained? If the silencing defense response is diverted from TSWV under such conditions, the TSWV accumulation level may be higher in mixed infection than in single infection. Should be clarified.
- b) In single infection, most TSWV-specific sRNAs are derived from the S component of the TSWV genome, whereas in mixed infection, they are derived from the M component of the TSWV genome. Do the authors have a hypothesis to explain this?
- c) Under the conditions of viral infection, the predominant size of endogenous (plant-specific) sRNAs is shifted from 24 to 21/22 nucleotides. Has this been described before in the literature? What might be the mechanism?
- d) Under mixed infection conditions, the authors observe that TSWV-derived vsRNAs are shifted from predominantly DCL2-produced to predominantly DCL4-produced, while no such shift is observed for INSV-derived vsRNAs. How could this be?
Minor point
Lines 284-285. In mock inoculated plants, 60.8% and 62.5% of sRNAs were of host origin (Figure 3).
What was the origin of the remaining sRNAs?
Author Response
This is an interesting paper describing the interaction of two orthotospoviruses, TSWV and INSV, in N. benthamiana plants. The authors show that TSWV and INSV exhibit antagonistic interactions under mixed infection conditions. Furthermore, using small RNA analyses in virus-infected plants, the authors show that these interactions are controlled by the host RNA silencing machinery, which is likely to differentially process TSWV and INSV genomic RNAs in mixed infections. The results are novel and definitely worth publishing in Viruses. However, I have some concerns that should be addressed before the paper can be accepted for publication.
Main concerns
- Some data are presented in supplementary figures and tables; however, the supplementary materials have not been made available to the reviewer. Therefore, an additional round of review is required after all data, including the supplementary materials, are made available.
Response: Supplementary files were provided in the original submission and were found by other reviewers, so we are sorry that they were not available to all. Please find them in the supplementary documents, and if not, please contact the editorial staff to let them know of the problem.
- In the Results and Discussion section, the discussion is provided in a minimalistic manner, so that many presented results, which are quite important, are not discussed at all. Such discussion should be included in the revised version of the paper.
Response: We thank the reviewer for this valuable suggestion. We have now substantially revised and added discussion points throughout the manuscript.
- Under mixed infection conditions, the amount of TSWV-specific sRNA was significantly lower than in TSWV-infected plants. How can this be explained? If the silencing defense response is diverted from TSWV under such conditions, the TSWV accumulation level may be higher in mixed infection than in single infection. Should be clarified.
Response: Thanks for this observation and suggestion. In our study, mixed infected plants consistently showed lower viral titers, measured by ELISA. We have found the similar outcomes by qPCR and sRNAs sequencing, where TSWV vsRNAs were lower than the single infection and INSV vsRNAs in mixed infection. We believe, reduction in TSWV vsRNAs in mixed infection reflects a true biological antagonism which could be due to the antagonistic interaction between INSV and TSWV in mixed infection, or elevated antagonistic response from the host, especially toward TSWV. We have revised it (line 313-317).
- In single infection, most TSWV-specific sRNAs are derived from the S component of the TSWV genome, whereas in mixed infection, they are derived from the M component of the TSWV genome. Do the authors have a hypothesis to explain this?
Response: We have addressed this concern in line 320-326. While we do not know the exact mechanism, this TSWV-specific change suggests a virus-virus interaction and/or a unique host-virus dynamic, or a difference in RNA folding and availability to the RNAi machinery.
- Under the conditions of viral infection, the predominant size of endogenous (plant-specific) sRNAs is shifted from 24 to 21/22 nucleotides. Has this been described before in the literature? What might be the mechanism?
Response: We thank the reviewer for this valuable suggestion. We have now added a whole section (section 3.5, line 348-361) addressing this concern.
- Under mixed infection conditions, the authors observe that TSWV-derived vsRNAs are shifted from predominantly DCL2-produced to predominantly DCL4-produced, while no such shift is observed for INSV-derived vsRNAs. How could this be?
Response: Thanks for the comment. We have observed TSWV-derived vsRNAs in single infection are shifted from predominantly DCL2-produced to both DCL2 and DCL4-produced. We did not see this change in case of INSV. While these changes are suggested by the size of the sRNA, additional experiments to verify these results should be performed. We made sure to emphasize this in the text (section 3.4).
Minor point
Lines 284-285. In mock inoculated plants, 60.8% and 62.5% of sRNAs were of host origin (Figure 3). What was the origin of the remaining sRNAs?
Response: Thank you for noting this. The remaining sRNAs not mapped to the host genome in mock-inoculated plants likely represent unannotated plant sequences, endophytic contaminants, or degradation products that do not align to the genome under our mapping criteria. We have now clarified this point in the section 3.3 (line 302-305).
Round 2
Reviewer 1 Report (New Reviewer)
Comments and Suggestions for Authors
The authors have revised the MS according to my comments. I recommand to accept it now.
This manuscript is a resubmission of an earlier submission. The following is a list of the peer review reports and author responses from that submission.
Round 1
Reviewer 1 Report
Comments and Suggestions for Authors
In the manuscript entitled “Small RNA Analysis of Virus-virus Interaction between Two Orthotospoviruses”, the authors confirmed that INSV and TSWV have an antagonistic relationship in N. benthamiana,and further determined small RNA (sRNA) profiles among the single or mixed infection, particularly INSV vsRNA and miRNA accumulation didn’t change much between single and mixed infections, but TSWV showed marked differences during mixed infections. The results are very interesting. However, the abstract and conclusions are not perfect and should be revised.
Comments raised:
1. Abstract should be described more results.
2. Conclusions should be revised.
3. Lines 127 and 132, please add full stop after references 27 and 39.
4. Line 247, please leave a blank space before “n=5”.
5. Some references is not consistent in format, for instance, “Molecular plant pathology” in reference 14 and “Molecular Plant Pathology” in 15.
Reviewer 2 Report
Comments and Suggestions for Authors
Overall evaluation:
Viral synergy is commonly existed in the natural patho-systems. Fully understanding of the mechanism of how two different viruses utilize the relative limited resources to achieved the co-robust infection of the same plant is essential for optimized disease management. Tomato spotted wilt virus (TSWV) and impatiens necrotic spot virus (INSV) are two important plant viruses that occurring on the tomato and impatient, which cause big threat to their production. In this study, the author proposed that the siRNA deep sequencing of the INSV-infected plant was not done previously, and the author expected to recover the relationship of the TSWV and INSV in co-infected plant by this sequencing. However, there are still many questions should be clarified. First of all, author think that the relationship of the TSWV and INSV was antagonistic in co-infection, which should be clarified by more inoculation experiments, such as using more inoculation methods (using purified virion, crude extract, or agrobacterium-based infiltration), the time of inoculation, and treatments of these two viruses before inoculation. In the next experiment, the author found that there was no super infection exclusion between TSWV and INSV. I wonder how does virus get this outcome. Overall, I did not think the siRNA deep sequencing data could well explain the antagonistic relationship between TSWV and INSV in plant, and there lacking solid experimental data to support the author’s point. I recommend to accept the MS after big revisions, reorganizations, resequencing using suitable samples, and solid experiment.
Major comments:
1. In treatment 2b shown in Table 1, comparing with the inoculum of 100% INSV + 100% TSWV, the TSWV infection rata were increased with inoculum of 100% INSV + 80% TSWV (The first two row, from 0% to 20%). In addition, in the third row to the fifth row, the results showed that the TSWV infection rata were not altered (from 20% to 20%). I think this is not convincing. As general, the higher contents of the TSWV in the inoculum always means higher incidence of the TSWV in the inoculated plants. How to explain this?
2. I also have a suggestion, that is to adjust the Table 1, which the top shows the Treatments, INSV infected, TSWV infected, and mixed-infected. Live the Treatment 1a, 2a, 1b, 2b as four individual rows following the Treatments.
3. As the author predicted, the antagonistic relationship between TSWV and INSV in plant, the interaction occurred in the cellular and molecular levels. Hence, the ELISA showed used to quantify the virus’s titer of the TSWV and INSV in the same cell or the same plant. All analyses showed based on the TSWV and INSV mixed infected plants. The ELISA data were also not convincing, such as, the bar is too large (it closed to 0.2), and I think there should have a statistic boxplot which contain results of multiple TSWV and INSV co-infected plants.
4. In Figure 2, the results showed that the total siRNA reads were severely decreased in TSWV and INSV co-infected plant than that of the TSWV or INSV single virus infected plant. Hence, i think the sequencing samples of the co-infected were not suitable, the titer of these two viruses in the co-infected plant was low. Hence, the obtained siRNA data of TSWV was very little compared to the single TSWV infection. Which was also evidenced by the results in Figure 3, that the mapping siRNA number of TSWV could achieve 40-thousand, while mapping siRNA number of in the co-infection sample only 2-thousand. This is a big difference, and suggested that the sequencing samples of the co-infected were not suitable.
5. In Figure 6, the PC assays, the results also showed that two repeats of sample that INSV or TSWV single infected plant were not well, and the dot of the INSV-infected sample and the dot of the TSWV- infected sample were closed to each other. Which indicating that samples for sequencing were not very well, or the repeatability of siRNA sequencing were not well.
6. The author should focus on the expression levels of the components of plant anti-viral RNA silencing, and using the obtained siRNA data to co-related with the expression levels, and find which target gene was determined the observed antagonistic relationship between TSWV and INSV in plant.
Reviewer 3 Report
Comments and Suggestions for Authors
The manuscript written by Kaixi Zhao et al, explore s the interaction between two orthotospovirus Tomato spotted wilt virus (TSWV) and impatiens necrotic spot virus (INSV) assessing the difference in transmission percentage, between individual infections of each virus and mixed infection (TSWV+ INSV), titer of the virus (DAS-ELISA), qPCR and sRNA sequencing. demonstrating that INSV and TSWV have an antagonistic relationship. The study is well conducted and the article well written, but some corrections should be considered.
1- The writing of the abstract can be improved
2- Lines 14-15 Please check the names of the viruses: Tomato spotted wilt virus (TSWV) and impatiens necrotic spot virus (INSV) not. Tomato spotted wilt orthotospovirus or impatiens necrotic spot orthotospovirus
3- Some Keywords are already used in the title I suggest to include Super infection exlusion
4- Line 41: This is likely triggered by RNAi interference (RNAi), I think it should be RNA instead of RNAi."
5- Line 65: severe strains of the same virus [15](15). the number of the reference is repited
6- Line 176 virus titer quantification by ELISA please explain how was the virus titer quantified, was it only by the OD405 lecture?
7- Line 190: were preformed using primers from [22] it would be better were preformed using primers discribed by Zhao, et al [22]
8- Line 225: its titer even in plants where TSWV was not detected by ELISA, Please clarify how the ELISA cut-off point was considered, to differentiate between healthy and diseased samples
9- Lines 252-256: the percentage of mixed infection is indicated first and then the number of plants with mixed or simple infections. I think that everything should be indicated as a percentage of infection
10- Line 574 bance, V.B. Replication of Potato Virus, has to be corrected Vance
11- Line 597 Plant Immune Receptor Adopts a Two-Step Recognition Mechanism to Enhance Viral EffectoráPerception. Delete de a Viral Effector Perception
12- Line 634 Please complete de Reference: Martinez-Ochoa, N, Csinos, A, Webster, T, Bertrand,P. Occurrence of mixed infections of tomato spotted wilt virus (TSWV) and impatiens necrotic spot virus (INSV) in weeds around tobacco fields in Georgia Available online:https://www.ars.usda.gov/research/publications/publication/?seqNo115=147365 (accessed on 12 April 2023).

In general the article is well written, but The writing of the abstract can be improved
Reviewer 4 Report
Comments and Suggestions for Authors
This is an interesting study showing a high level, but incomplete, repression of TSWV by INSV in co-infected plants. The siRNA data confirms the dominance of INSV. However, the experiments were done in broad strokes, leaving many questions unanswered. Additionally, the writing of the manuscript could be substantially improved. Some examples are provided below.
Major issues:
Table 1: Please clarify: (i) what do “Treatment 1a” and “Treatment 2a” mean? (ii) what do the percentages behind the virus names mean? I am guessing the “1a” column were controls, with the virus titers gradually reduced. If I am right, then it would be much less confusing to list the “2a” treatments directly below the “1a” treatments. Furthermore, they really should not be named as 1a and 2a – they were probably done at the same time, correct? Also for diluted inoculums, do not use percentage as it is easily confused with the diseased plant percentages – I would simply use something like 4:5 diluted, 3:5 diluted, 2:5 diluted, and 1:5 diluted).
Same goes for 1b and 2b.
The data revealed a one-way repression against TSWV by INSV. Did INSV repress TSWV in thrips? At what time post inoculation were the ELISA tests done?
Ln249-259, please clarify (i) what was the time lag between two inoculations; (ii) at what time point were the plants tested for the presence of the viruses; (iii) whether the plants were examined at multiple time points.
Comments on the Quality of English Language
Examples of writing issues:
Ln16-17, delete “more”. Also state in which host(s). Delete “and the INSV host range ……” – this information is not critical for Abstract, hence can be provided later in Introduction.
Ln18, are these new strains? If yes, state so (e.g. we have isolated a new strain of INSV and also a new strain of TSWV). If not, then you need to emphasize their co-existence by moving that information to the beginning of the sentence.
Ln19, provide the name of the vector(s). Remove the coma after “determine”.
Ln20, if the antagonism occurs both in transmitting vectors and in plants, then you need to add this info in the line above. If the “in planta” antagonism is a new discovery of this paper, then state so clearly.
Ln22, I suspect it should read “INSV alters plant responses to TSWV by perturbing the accumulation of TSWV-specific small interfering RNAs”?
Ln32, change to “…… that subdued the antiviral RNA silencing targeting PVX”.
Ln34, “folds” should be “fold”. Similar example: “this cup costs 10 dollars, but it is a 10 dollar cup”.
Ln44, change “happens” to “was also observed in xx (tomato? Tobacco?) plants infected with TSWV and ToCV”. You already indicated in the previous paragraph that synergy is a plant response, thus cannot occur between viruses independent of plants.
Ln46, after “orthotospoviruses” add “mediated by the Sw-5 resistance gene”; “be overcome by pre-infection of an unrelated virus”. End the sentence here, restart the next sentence as a new sentence. “It was reported that …….”
Ln47, “are” should be “were”; after ToCV, “they became susceptible to TSWV inoculation merely 10 days later”.
The Introduction should be substantially shortened, with emphasis on antagonistic interactions, which is the focus of the current study.
Ln206, add the host! As pointed out by the authors themselves, virus-virus interaction is only relevant in shared hosts.